# Abdominal Pain in Inflammatory Bowel Diseases: A Clinical Challenge

**DOI:** 10.3390/jcm11154269

**Published:** 2022-07-22

**Authors:** Pauline Wils, Bénédicte Caron, Ferdinando D’Amico, Silvio Danese, Laurent Peyrin-Biroulet

**Affiliations:** 1Department of Gastroenterology, Claude Huriez Hospital, University of Lille, F-59000 Lille, France; 2Department of Gastroenterology, University of Lorraine, CHRU-Nancy, F-54000 Nancy, France; caron.benedicte@hotmail.fr (B.C.); peyrinbiroulet@gmail.com (L.P.-B.); 3Department of Gastroenterology, University of Lorraine, Inserm, NGERE, F-54000 Nancy, France; 4Gastroenterology and Endoscopy, IRCCS Ospedale San Raffaele, 20132 Milan, Italy; damico_ferdinando@libero.it (F.D.); sdanese@hotmail.com (S.D.); 5Department of Biomedical Sciences, Humanitas University, 20090 Milan, Italy

**Keywords:** inflammatory bowel disease, abdominal pain, quality of life

## Abstract

Up to 60% of inflammatory bowel disease (IBD) patients experience abdominal pain in their lifetime regardless of disease activity. Pain negatively affects different areas of daily life and particularly impacts the quality of life of IBD patients. This review provides a comprehensive overview of the multifactorial etiology implicated in the chronic abdominal pain of IBD patients including peripheral sensitization by inflammation, coexistent irritable bowel syndrome, visceral hypersensitivity, alteration of the brain–gut axis, and the multiple factors contributing to pain persistence. Despite the optimal management of intestinal inflammation, chronic abdominal pain can persist, and pharmacological and non-pharmacological approaches are necessary. Integrating psychological support in care models in IBD could decrease disease burden and health care costs. Consequently, a multidisciplinary approach similar to that used for other chronic pain conditions should be recommended.

## 1. Introduction

Updated STRIDE recommendations (STRIDE II) confirm abdominal pain as a relevant target in patients with Crohn’s disease (CD) [1,2]. Despite the current treat-to-target strategies to improve patient-reported outcomes (PRO), abdominal pain is still a frequent symptom in patients with inflammatory bowel disease (IBD) [3]. Up to 60% of patients with CD or ulcerative colitis (UC) experience chronic abdominal pain leading to impacts on daily life and increased psychosocial burdens [4].

Abdominal pain pathogenesis and its perception have multifactorial etiology and can be modulated by many factors in patients with IBD. Pain can result from ongoing inflammation or coexistent functional disorders like irritable bowel syndrome (IBS) [5]. Literature data demonstrated that in IBD, inflammation sensitizes primary afferent neurons leading to chronic abdominal pain [6,7]. Moreover, alterations of the brain–gut interactions can modulate the perception of pain, contributing to its occurrence [8].

Unfortunately, pain is still under treated (24% of IBD patients received no treatment for pain in a recent study) [3], and specific treatments for pain management in IBD are poor. Pain control has been identified as one of the top 10 research treatment priorities by clinicians and IBD patients [9]. Opioids and cannabis are frequently used for pain control by IBD patients (14.7% and 17%) [10,11]. However, the chronic use of opioids exposes subjects to addiction and relevant side effects, particularly constipation (15% to 40%), confusion (at high doses), nausea and vomiting (10% to 40%), sedation (20% to 60% of patients receiving oral morphine for cancer pain), and, more rarely, respiratory depression [12]. The regular use of opioids is an independent predictor of mortality in IBD patients [13]. Therefore, increasing the knowledge on pain pathophysiology in IBD is crucial for a better therapeutic management of chronic abdominal pain. This article will review the etiological factors and consequences of abdominal pain in IBD patients, focusing on currently available pharmacological and non-pharmacological treatments.

## 2. Assessment and Consequences of Abdominal Pain in IBD

Abdominal pain can be categorized as acute or chronic based on duration. Chronic pain is defined as pain that occurs consistently for 3 months or intermittently for 6 months [14]. Abdominal pain in IBD patients can result from ongoing inflammation. Several indices developed to evaluate disease activity (Crohn’s disease activity index (CDAI) for CD or Lichtiger index for UC) include abdominal pain.

Pain is also a subjective experience, and a patient’s self-report of pain scale must be encouraged. There are a number of validated pain scales that measure pain intensity: the Visual Analog Scale (VAS) or Numeric Rating Scale (NRS) are the most numerical rating scales for pain measurement used in studies (ranging from 0 “no pain” to 10 for NRS or to 100 mm for VAS corresponding to “worst pain imaginable”), but they are not specific to abdominal pain in IBD [15]. The irritable bowel syndrome–Symptom Severity Scale (IBS–SSS) integrates abdominal pain; however, it is not validated for IBD patients. In addition to these numerical scores based only on intensity measurement, there exist more detailed scales. The McGill pain questionnaire solicits qualitative descriptions of pain (15 descriptors in the short form) [16]. The Brief Pain Inventory assesses the intensity (severity scale) and location of pain as well as its impact on patients’ daily functioning (interference scale) [17]. In practice, asking patients to use a scale of 0–10 concerning pain severity and the impacts of pain on their lives can be sufficient.

Pain in IBD can interfere with different areas of daily life. IBD has a substantial impact on quality of life (QOL) during periods of both active and inactive disease, including considerable personal, emotional, and social burdens and work-related difficulties [18,19]. In IBD, IBS-like symptoms (such as abdominal pain) are associated with poor QOL [20]. An association between pain and health-related QOL is well-known in chronic diseases that also exists in IBD [21,22]. Symptoms of active disease have been reported in multinational questionnaire-based studies to have substantial impacts on health-related QOL [23]. A longitudinal study examining the impacts of persistent gastrointestinal symptoms (such as abdominal pain) in IBD patients revealed higher anxiety and depression and lower QOL compared with those without symptoms [24]. In a large pan-European survey study including 4670 IBD patients, 87% of the respondents had experienced abdominal pain at least one day a week during their last flare-up, 34% daily, and 62% between flares. In this study, abdominal pain and fatigue were the most common reason given for work absenteeism (46% and 51%, respectively) [25]. A recent national Swiss cohort study enrolling 2152 IBD patients showed that pain was a longstanding problem (>5 years) for a relevant proportion of patients (49% of UC and 55% of CD patients). In this study, the abdomen was the most frequent location of pain (59.5%), with a significant reduction of QOL compared with subjects without pain (38 vs. 77; (−100 very bad; 100 very good) *p* < 0.0001) [3]. Moreover, an another study on 334 IBD patients reported that 87.9% of them had pain and showed significantly reduced QOL compared with healthy controls (*p* = 0.0001) [21].

Abdominal pain has direct and indirect health care costs including reduced employment and productivity. There is growing support that opioid use was among the top factors associated with high health care utilization and costs for patients with IBD [26].

## 3. Mechanisms of Chronic Abdominal Pain

The heterogeneity in the perceptions of chronic abdominal pain by IBD patients can be explained by different mechanisms with multiple contributing factors that are summarized in Figure 1.

### 3.1. Direct Effect of Inflammation

A majority of IBD patients experience abdominal pain during acute flares [27]. Acute pain may reveal complications: partial or complete gut obstruction, fistulas, or abscesses. It is commonly assumed that abdominal pain results directly from inflammation and may be correlated with the degree of disease activity [28]. However, in many IBD patients, there is a discrepancy between the absence of objective inflammation on endoscopic investigations or biomarkers and the intensity of pain perception. Around 20% of patients in complete endoscopic remission experienced pain [27]. The mechanism of the persistence of pain despite a lack of objective inflammation is unclear and can be controversially explained as irritable bowel syndrome [29]. Other complications may arise during IBD management and should be investigated in the absence of inflammation: adherences, small intestinal bacterial overgrowth, and colorectal cancer [7].

### 3.2. Peripheral and Central Pain Dysregulation

The mechanisms of chronic abdominal pain in IBD are complex and involve the enteric nervous system (ENS), the central nervous system (CNS), the gastro-intestinal immune system, and the intestinal barrier, leading to the dysregulation of brain–gut interactions [30,31]. The ENS is the intrinsic innervation of the bowel, which controls gastro-intestinal functions independent of the CNS (like motility, secretion). The ENS, located in the wall of the gastrointestinal tract, has a pivotal role, receiving input from the CNS and the autonomic nervous system and interacting with the immune system of the gut [32].

In IBD, chronic visceral pain can result from peripheral sensitization by inflammation. During inflammation, a multitude of proinflammatory cytokines (interleukin (IL)-1β, IL-6, tumor necrosis factor (TNF)-alpha), mediators (for example, substance P, 5-hydroxytryptamine), and neuropeptides (including substance P, nerve grow factor, calcitonin gene-related peptide) are release by tissue damage [32,33] or immune cells (mast cells or leukocytes). These molecules activate the visceral afferent neurons and sensitize nociceptive receptors [27,34]. Pain is also driven by the activation of visceral nociceptors in response to potentially damaging stimuli on the gut (for example distension); these depolarize the nerve terminal and transmit pain information to the CNS.

In IBD, inflammation can generate nociceptive messages to the CNS, which prolongs or amplifies the sensitization of visceral afferents, contributing to chronic abdominal pain (central sensitization) [35]. Moreover, the descending pathway from the CNS modulates pain transmission by increasing input (facilitation) or decreasing input (inhibition). An imbalance in the descending pain modulation to facilitation can lead to chronic pain. These mechanisms are implicated in visceral hypersensitivity.

Abnormalities of the CNS and more particularly in the prefrontal and limbic regions are also implicated in the amplification of perceptive pain and in humans’ incapacity to manage pain [36]. Moreover, brain regions associated with emotional regulation and their dysfunction contribute to increasing the risks of depression and anxiety in IBD patients [37,38].

### 3.3. Overlap between IBD and IBS

IBS is a functional gastrointestinal disorder that shares common symptoms with IBD: recurrent abdominal pain and changes in stool frequency [29]. IBS affects 11% of the general population [39] and approximately 40% of patients with active or quiescent IBD [40]. In a recent meta-analysis, symptoms compatible with IBS concerning IBD patients in endoscopic or histologic remission are reported in approximately 25% of cases and significantly more in CD than UC patients (34.9% vs. 29.1%; OR 1.58; 95% CI 1.27–1.98) [29]. In clinical practice, it is difficult to differentiate functional symptoms from ongoing active CD or UC due to their frequent prevalence.

The pathophysiology of pain in IBS is multifactorial, including a dysregulation of brain-gut interactions and an increased activation of the brain regions involved in emotional and pain control [41]. Moreover, psychological disorders associated with pain, like depression and anxiety, are reported both in IBD and IBS patients. In a recent meta-analysis, IBS patients had higher rates of depression and anxiety than general populations (mean difference respectively: 9.87 CI95 (6.3–13.4) and 8.10 CI95 (3.7–12.5), *p* < 0.01) [42].

An abnormal microbiome may be implicated in the development of IBS-type symptoms and abdominal pain. Dysbiosis is associated with IBD pathogenesis [43], suggesting—in comparison with IBS—a role of gut microbiota in visceral hypersensitivity, but data remain scarce. A previous study on the fecal microbiome in IBD patients with IBS-like symptoms did not find any difference in the bacterial abundance or diversity between patients who reported symptoms and those who did not [44]. Then, Pérez-Berezo et al. reported that an analgesic lipopeptide related to GABA was produced by the probiotic Escherichia coli strain Nissle 1917 [45]. Related to this, probiotics are a presumed therapeutic option in IBD–IBS patients for improving pain.

Small bowel intestinal overgrowth, frequently revealed by chronic abdominal pain and other gastrointestinal symptoms, may be associated with gut microbial dysbiosis and influenced by many factors in patients with IBD: resection of the ileocecal valve, intestinal motility disorders, or complications such as fistulas or stenoses. Overlap between small bowel intestinal overgrowth and IBD was not uncommon and was reported in more than 20% of IBD patients in a recent meta-analysis [46].

### 3.4. Impact of Psychological and Social Factors on Pain

CD and UC are associated with significant psychological comorbidity in both adult and pediatric patients [47]. Anxiety and depression are the most frequent mood disorders affecting IBD patients [48]. A recent systematic review and meta-analysis of mood disorders in IBD patients reported a prevalence of anxiety in 32.1% of patients and a prevalence of depression symptoms in 25.2% in pooled studies [49]. A bi-directional interaction was also suggested between IBD activity and psychological disorders (anxiety or depression contributing to the progression of IBD and IBD affecting psychological health) in a prospective longitudinal 2-year study. In this study of 405 patients, IBD activity and anxiety/depression scores were reported at baseline and after 2 years of follow-up: IBD activity at baseline was associated with a higher anxiety score at the end of follow-up (HR: 5.77; 95% CI, 1.89–17.7), and an abnormal anxiety score in quiescent IBD patients at baseline was associated with later IBD flare-up [50]. Chronic gut inflammation may impact psychological health. Of note, the administration of cytokines like TNF-alpha, IL-1β, and IL-6 in healthy volunteers can modify central neurotransmitter release and behavior leading to depressed mood [51,52]. Mood disorders have been shown to amplify symptom severity, particularly abdominal pain perception [53]. In pediatric and adolescent patients with IBD, depression and child pain catastrophizing may increase the sensitivity to abdominal pain and pain impact [54]. Moreover, in IBD, stress can lead to abdominal pain, affecting thresholds for pain perception and visceral hypersensitivity [55]. In accordance with this hypothesis, Schirbel et al. reported that mental stress intensified pain by 38% in IBD patients [21].

Several psychological and social components contribute to pain in IBD patients. Depression and anxiety followed by perceived stress and pain catastrophizing were positively associated with increased pain [56]. As a vicious circle, increased abdominal pain was associated with mood disorder (OR = 5.76, CI 95%:1.39–23.89) [56]. On the other hand, perceived social support and internal locus of control were considered protective psychosocial factors and associated with less pain. A recent study including 297 IBD patients confirmed these data. Female gender, smoking, surgery, and steroids were associated with greater pain severity. Interestingly, psychosocial factors such as depression, catastrophizing, fear avoidance, lower self-efficacy, and worse mental well-being were associated with pain-related interference [57].

### 3.5. Genetic Factors

Individual variation in pain perception is influenced by environmental but also genetic factors [58]. Genetic predisposition is implicated in IBD pathogenesis, and more than 240 single-nucleotide polymorphisms (SNPs) are associated with the risk for IBD [59]. A genetic link between IBD and IBS has been postulated after the identification of increased transient receptor vanilloid type 1 (TRPV1) nerve fibers. These are implicated in visceral hypersensitivity and in quiescent IBD–IBS patients with a correlation to pain severity [60]. Recently, genome-wide association studies have identified other SNPs associated with IBS [61]. Similarly, another study showed that two IBS-associated SNPs were associated with maximal abdominal pain in UC patients [62], suggesting the involvement of genetic factors in these patients.

## 4. Pain Management

Many IBD therapies (anti-TNF, anti-integrin, anti-interleukin, small molecules) demonstrated their efficacy in multiple randomized controlled trials (RCTs). Nevertheless, pain improvement was not considered a key endpoint in the majority of studies even though it is an essential outcome for patients. More recently, endpoints have evolved, and abdominal pain improvement is considered a secondary endpoint in RCTs evaluating jak inhibitors (tofacitinib and upadacitinib) versus placebo in UC [63,64]. Control of disease activity in IBD is the first step in the management of abdominal pain, but despite an optimal management of intestinal inflammation, chronic abdominal pain can persist, and pharmacological and non-pharmacological approaches can be necessary [6,30,65,66]. Recently, the American Gastroenterological Association consensus has described the key principles in the management of functional gastrointestinal symptoms in IBD patients [5].

Figure 2 summarizes in an algorithm a sequential approach for abdominal pain management. The treatment of acute pain depends on pain intensity (assessing by VAS or NRS). Simple analgesics are often proposed in the first step, and in the case of severe pain (VAS > 55 mm or NRS > 7), opioids can be temporarily prescribed. An evaluation of IBD activity is recommended, and IBD therapies were modified accordingly. In quiescent diseases, a low FODMAPs diet may be an interesting option in IBD–IBS patients, psychological interventions may be proposed in patients with associated mood disorders, and adjuvant drugs with antidepressant and analgesic effects can be useful in most patients with chronic pain. Integrating psychological support in care models in IBD could decrease disease burden and health care costs. Consequently, a multidisciplinary approach with a mental health professional for assessing the degree of anxiety and/or depression and evaluating the probability of responding to specific medications or psychological therapies is recommended for IBD patients.

### 4.1. Current Pharmacological Options

No therapies have yet proven to be effective in managing IBD-related pain.

Nonsteroidal anti-inflammatory drugs are not recommended for long-term use because of the risk of relapse or mucosal injury. Although there is no definite evidence in IBD patients, antispasmodics are commonly used in mild pain [67]. However, antispasmodics and simple antalgics are often not sufficient, leading to the prescription of opioids. Despite their potential side effects, a high proportion of IBD patients received opioids for chronic abdominal pain [10]. Notably, in a large epidemiological study, approximately 5% of IBD patients were heavy users of opioids within 10 years of diagnosis [13]. However, the chronic use of opioids should be avoided in IBD patients due to the risks of dependence, hyperalgesia, narcotic bowel syndrome [68], and premature mortality in heavy users [5,69].

Alternative drugs to opioids should be encouraged [6,7,65,66]. Antidepressant and anxiolytic medications may be an interesting option for abdominal pain especially if there are coexisting mood disorders. In particular, a retrospective cohort study involving 81 patients with IBD revealed that tricyclic antidepressants (amitriptyline, nortriptyline, or desipramine) improved gastrointestinal symptoms (not abdominal pain specifically) [70]. An open-label preliminary study on selective serotonin reuptake inhibitors (citalopram, paroxetine, fluoxetine, and escitalopram) in psychiatric disorders suggested the efficacy of paroxetine on abdominal pain in eight IBD patients [71]. Serotonin–norepinephrine reuptake inhibitors (SNRIs) were effective for the control of chronic pain syndromes in IBS patients. Interestingly, a large Canadian cohort study of 403,665 patients with new-onset depression investigated the impacts of depression and antidepressant therapies on the development of IBD. Selective serotonin reuptake inhibitors and tricyclic antidepressants were associated with a reduced risk of developing both CD and UC, and SNRIs were protective against UC [72]. Nevertheless, these treatments are not devoid of side effects, (nausea, dry mouth, muscle spasms, constipation), bringing an additional burden to patients. Other studies suggest a potential abdominal pain-relieving effect of gabapentin and pregabalin (usually prescribed for neuropathic pain), but these have not been investigated in IBD patients [6]. Other interventions have been associated with a reduction of abdominal pain [66]. In a randomized controlled trial evaluating the efficacy of antibiotics on 29 CD patients with small intestinal bacterial overgrowth, the antibiotics metronidazole and ciprofloxacin improved abdominal pain in 50% and 43% of cases, respectively [73]. Rifaximin, a non-absorbed antibiotic, was an effective treatment of small intestinal bacterial overgrowth, improving symptoms in 67% of patients in a recent meta-analysis [74]; it could be an option for IBD patients with small intestinal bacterial overgrowth. Furthermore, restoring the microbial flora seems to be an interesting option in the management of IBS. Some probiotics have demonstrated their effectiveness in improving symptoms, including abdominal pain for patients with IBS (for example L. Plantarum 299v, B. Bifidum MIMBb75) [75,76]. The use of probiotics in IBD could be a therapeutic option to improve dysbiosis but requires further studies. Another randomized study demonstrated the efficacy of transdermal nicotine patches in 72 active UC patients versus placebo: 49% in the nicotine group had complete remission compared with 24% in the placebo group (*p* = 0.03), with less abdominal pain in the nicotine group (*p* = 0.05) [77]. A small trial evaluating loperamide oxide in CD patients with chronic diarrhea indicated an improvement in abdominal pain after one week of treatment [78]. Previous studies reported that around 10% of IBD patients use cannabis and derivatives to improve symptoms [79]. Patients developed a growing interest in the potential therapeutic effect of cannabis on IBD-related symptoms and IBD course. Data regarding the inflammatory effects of cannabis and derivatives are still limited: small studies reported an improvement of several IBD-related symptoms, including abdominal pain, but without any significant improvement of inflammation markers or disease course [11,80,81]. A phase 2a trial evaluating the efficacy and tolerability of cannabidiol in steroid-dependent CD patients is still ongoing [82]. Table 1 details pharmacological options in IBD patients with chronic abdominal pain.

### 4.2. Non-Pharmacological Interventions

#### 4.2.1. Dietary Measures

The presence of indigestible carbohydrates or osmotic molecules in the luminal gut may result in high fermentation, the production of gas and short chain fatty acids, and distention and could participate in IBS-like symptoms such as abdominal pain in IBD patients [87]. Lactose and fructose malabsorption are frequent in CD patients, an estimated 42% and 61% of patients, respectively, in a previous study [88] and could be associated with abdominal pain. Several dietary measures including lactose-reduced or low fermentable oligosaccharides, disaccharides, monosaccharides, and polyols (FODMAPs) diets may improve functional gastrointestinal symptoms [87]. Despite the low grade of evidence, the low-FODMAP diet proved to be effective for the IBS-like symptoms of IBD patients in remission [83]. In addition, a randomized controlled cross-over trial reported an exacerbation of functional gastrointestinal symptoms in quiescent IBD patients exposed to FODMAPs [89]. However, considering the risk of compromised nutritional status, the use of restrictive diets in IBD patients in remission should be carefully supervised by a dietician.

#### 4.2.2. Psychological Approaches

Psychosocial factors impact the quality of life of IBD patients and often can benefit from non-pharmacological approaches. Notably, the consequent association between pain and depression, stress, or anxiety suggests that the optimal management of these mood disorders may improve pain levels and QOL. In the IBD field, psychological interventions, particularly cognitive behavioral therapy (CBT), have been found as valid approaches for functional gastrointestinal symptoms and pain [7,66]. Coping skill training, another psychological and educational intervention, developed to increase individuals’ ability to manage uncomfortable or anxiety-provoking situations, improved somatic symptoms in IBD adolescents but did not significantly affect pain [84]. Notably, psychological interventions were associated with clinical improvement in pediatric and adult IBD patients with anxiety or depression [85,90,91]. However, another randomized controlled trial evaluating the impact of a 10-week cognitive behavioral therapy in adult IBD patients did not find an influence of CBT in the course of IBD over 24 months [92]. Another proposed option is hypnotherapy, a technique used to create a state of focused attention and help to gain control over undesired behaviors or treat stress with suggestions for relaxation. Hypnotherapy has been associated with positive outcomes for different chronic-pain conditions (e.g., cancer, low-back pain, arthritis pain). For IBD, a prospective study demonstrated a significant effect of gut-directed hypnotherapy on prolonging clinical remission in 54 patients with quiescent UC [85]. There is also consistent evidence for a contribution of stress in the IBD disease course [93]. For this reason, a stress management program including 45 CD patients was proposed. A significant reduction in abdominal pain was detected in subjects who received training in self-directed and therapist-led stress management [86]. Acupuncture, derived from traditional Chinese medicine, is a practice designed to rebalance a patient’s Qi. In IBD patients, acupuncture can be a complementary approach that helps reduce chronic abdominal pain [94].

## 5. Future Directions

Further therapeutic options are necessary. Among therapeutic options with preliminary data, selective blockade of transient receptor potential Vanilloid 4 (TRPV4) [65,95], and NOP (nociceptin/orphanin peptide receptor) agonists [96,97,98,99] could be new potential therapeutic options with possible anti-inflammatory and antinociceptive actions. A previous study indicates that TRPV4 localized in the gastrointestinal tract may play a role in mechanisms of defense in intestinal inflammation and that selective blockade of TRPV4 in animal model alleviates colitis and pain associated with the intestinal inflammation [100]. Preliminary data suggest a protective role of the nociceptin/orphanin FQ–NOP receptor system (N/OFQ ligand and its receptor nociceptin/orphanin peptide) in the pathogenesis of IBS–D and as a potential target for intestinal disorders [96]. NOP receptors share similarities with opioid receptors but with very low affinity to opioid ligands. NOP receptors are distributed in the central and peripheral nervous systems and also in the gastrointestinal tract, where they may interact with gastrointestinal functions like motility, secretion, or pain perception [97]. A NOP agonist was developed in a mouse model of IBD with possible anti-inflammatory and antinociceptive action [99].

Among new approaches to pain management, transcranial direct current stimulation acting on the central mechanisms of pain is currently being explored in IBD patients. It proved to be a relevant strategy for chronic abdominal pain in 20 IBD patients in a randomized controlled double-blind study [101].

However, progress is limited by insufficient understanding of the causes of chronic abdominal pain in IBD. To try to respond to this need, the Crohn’s & Colitis Foundation launched in 2021 the chronic pain in translational IBD research initiative focused both on the understanding of chronic abdominal pain and its clinical management in IBD [102]. Finally, educating gastroenterologists on the management of chronic pain including non-pharmacological treatments should improve access for patients with IBD to all available treatments.

## 6. Conclusions

Abdominal pain remains a common symptom in IBD patients with a negative impact on daily life. Currently, there are several options for treating visceral pain but not specifically for IBD. This review highlights first a new attention to pain recognized as a burden for patients and second the lack of consensus on managing long-term pain in IBD. The resolution of abdominal pain should be incorporated as an independent endpoint into clinical trials for optimizing IBD management. As chronic pain is complex and associated with emotional and social factors, a multidisciplinary approach involving psychologists, dieticians, pain therapists, and psychiatrists is recommended.

## Figures and Tables

**Figure 1 jcm-11-04269-f001:**
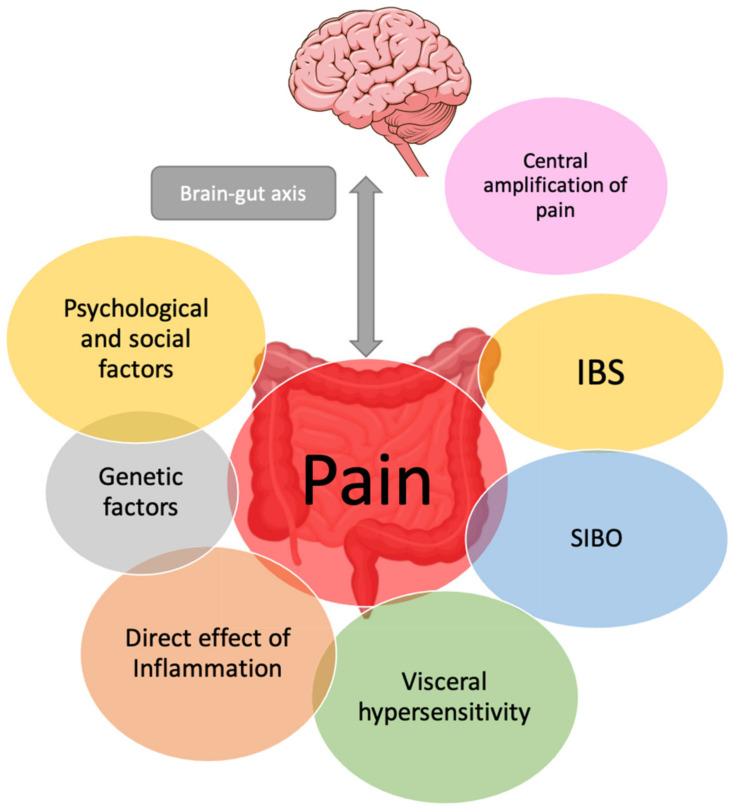
Proposed representation of mechanisms and multiple contributing factors implicated in abdominal pain in IBD: psychological and social factors, genetic factors, direct effect of inflammation, visceral hypersensitivity, co-existent IBS or central pain dysregulation. Abbreviations: IBS: irritable bowel syndrome; SIBO: Small intestinal bacterial overgrowth.

**Figure 2 jcm-11-04269-f002:**
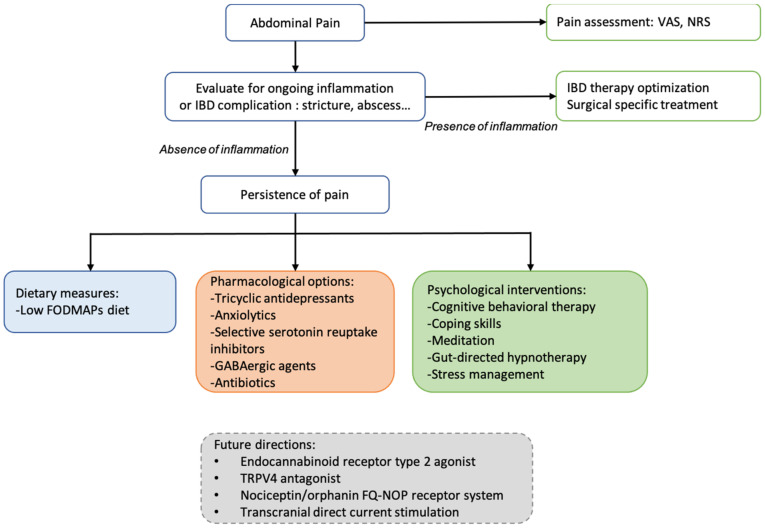
Proposed algorithm for pain management with pharmacological agents and non-pharmacological interventions available for improve chronic abdominal pain.

**Table 1 jcm-11-04269-t001:** Data supporting the efficacy of pharmacological and non-pharmacological interventions in IBD patients with chronic abdominal pain.

Treatment	Study Design	Study Intervention	Age (Year) Sex F	Number of Patients	Abdominal Pain Outcome
**Pharmacological treatment**					
tricyclic antidepressants (TCA) [70] (nortriptyline, amitriptyline, desipramine, doxepin)	Retrospective cohort study	IBD patients with inactive or mildly active disease and persistent gastrointestinal symptoms (median TCA dose: 25 mg (10–150 mg))	41.3 69%	58 CD/23 UC	TCA improved gastrointestinal symptoms in 59.3% of IBD patients (Likert score ≥ 2) Response was better in UC than in CD patients (1.86 ± 0.13 vs. 1.26 ± 0.11, respectively, *p* = 0.003)
Antibiotics: metronidazole or ciprofloxacin [73]	RCT	CD patients with small intestinal bacterial overgrowth (confirmed by hydrogen/methane breath and glucose tests) receiving metronidazole 250 mg t.d.s (group A) or ciprofloxacin 500 mg b.d (group B) for 10 days	39 41%	29 CD	Improvement of abdominal pain in 50% (group A) and 43% (group B) of cases
Transdermal nicotine patch [77]	Randomized double-bind study	Transdermal nicotine (5 or 15 mg) versus placebo in active UC patients; improvement of abdominal pain was a secondary outcome.	44 43%	72 UC	Abdominal pain rate on 0–2 scale at 6 weeks was at 0.3 inthe nicotine group and at 0.6 in the placebo group (*p* = 0.05)
Loperamide oxide [78]	Double-blind investigation	Loperamide 1 mg or placebo after passage of each unformed stool for one week	35 53%	34 CD	At one week, the investigator’s assessment of the change in abdominal pain was significant for loperamide oxide (*p* = 0.020) but not for placebo.
Cannabis [11]	Monocentric cohort	Consecutive patients with IBD who had used cannabis specifically for the treatment of IBD or its symptoms were compared with those who had not	36.6 50% (users)	303	17.6% of patients used cannabis to relieve symptoms associated with their IBD. Cannabis improved abdominal pain (83.9%), abdominal cramping (76.8%), joint pain (48.2%), and diarrhea (28.6%), although side effects were frequent.
**Dietary measures**					
Low-FODMAPs diet [83]	Retrospective telephone survey	IBD patients in remission Improvement of 5 points or more for gastrointestinal symptoms after dietary information on low-FODMAPs diet	48 39%	52 CD/20 UC	Approximately 70% of patients were adherent to the low-FODMAPs diet After 3 months, 56% had clinical improvement of abdominal pain (*p* < 0.02)
**Psychological approaches**					
Cognitive behavioral therapy [84]	RCT (CBT versus supportive nondirective therapy)	Evaluation of IBD activity (PCDAI and PUCAI) and depression in young patients (after 3-month course of CBT or supportive nondirective therapy	14.3 46% and 52%	161 CD and 56 UC	Compared with supportive non-directive therapy, CBT showed a greater reduction in IBD activity (*p* = 0.04); both psychotherapies decreased rate of depression scale
Gut-directed hypnotherapy [85]	RCT hypnotherapy (HPN) versus nondirective discussion	Patients received seven sessions of HPN or nondirective discussion. Evaluation of proportion of participants in each condition that had remained clinically asymptomatic through 52 weeks post treatment	38 54%	54 quiescent UC	68% versus 40% of patients maintaining remission for 1 year (*p* = 0.04)
Stress management program [86]	RCT stress management, self-directed stress management, or conventional medical treatment	CD patients considered in non-active stage of disease under sulfasalazine Evaluation of symptoms post-treatment	31.7 64%	45 CD	Significant decrease in abdominal pain in both stress management arms (14.2% and 6.6% versus 48%)

Abbreviations: TCA: tricyclic antidepressants; CD, Crohn’s disease; UC, ulcerative colitis; FODMAP: fermentable oligosaccharides, disaccharides, monosaccharides, and polyols; RCT: randomized controlled trial; CBT: cognitive behavioral therapy; PCDAI: Pediatric Crohn’s Disease Activity Index; PUCAI: Pediatric Ulcerative Colitis Activity Index; HPN: hypnotherapy.

## Data Availability

Not applicable.

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
