# Peer review of "Abdominal Pain in Inflammatory Bowel Diseases: A Clinical Challenge"

_jcm, 2022, doi:10.3390/jcm11154269_

Round 1
Reviewer 1 Report
This is a very well-written review paper concerning a relatively neglected topic in IBD.
The authors performed a very comprehensive summary of the current knowledge on different and multifactorial etiologies of abdominal pain in IBD. It is especially important for the clinicians to remember those etiologies not related to the activity of inflammation since the vast majority of therapies used in IBD are directed only against inflammation.
Namely, the authors are discussing the potential application of a low FODMAP diet in the treatment of pain in IBD. Since this dietary intervention is mainly used in SIBO, I would suggest also discussing this issue in the article. Is SIBO a potential other cause of abdominal pain in IBD (except of study by Castiglione et al.) ? What is the frequency of SIBO in IBD? Are there any data on the efficacy (in reducing abdominal symptoms in IBD) of other therapeutic interventions typical for SIBO (the data on the possible application of rifaximin in IBD?)?
Author Response
Lille, July 2022, 14th
Dear Editors and Reviewers,
We are pleased with the interest you put into analyzing our paper entitled “Abdominal Pain in Inflammatory Bowel Diseases: a clinical challenge”, and we are thankful for the quality of the comments and the rapid processing, which have allowed us to improve the quality of our work. The issues that you identified were analyzed and changes in the writing have been done accordingly. Reviewer questions and concerns are addressed point by point in the following section. All manuscript changes resulting from reviewer suggestions are embedded in the paper and highlighted (in red color) as required.
Reply for reviewer 1
Namely, the authors are discussing the potential application of a low FODMAP diet in the treatment of pain in IBD. Since this dietary intervention is mainly used in SIBO, I would suggest also discussing this issue in the article. Is SIBO a potential other cause of abdominal pain in IBD (except of study by Castiglione et al.) ? What is the frequency of SIBO in IBD? Are there any data on the efficacy (in reducing abdominal symptoms in IBD) of other therapeutic interventions typical for SIBO (the data on the possible application of rifaximin in IBD?)?
We thank the reviewer for this comment. The low FODMAP diet is used by some clinicians as an option in patients with small intestinal bacterial overgrowth particularly after antibiotics failure.
Small bowel intestinal overgrowth, frequently revealed by chronic abdominal pain, may be associated with gut microbial dysbiosis and influenced by many factors in patients with IBD: resection of the ileocecal valve, intestinal motility disorders or complications such as fistulas or stenoses. Overlap between SIBO and IBD was not uncommon and was reported in more than 20% of IBD patients in a recent meta-analysis.
Rifaximin, a non-absorbed antibiotic was an effective treatment of small intestinal bacterial overgrowth improving symptoms in 67% of patients in a recent meta-analysis and could be an option for IBD patients with small intestinal bacterial overgrowth.
As requested, these modifications appear in red color in the paragraph “overlap between IBD and IBS” on Page 4 and in the paragraph “Pain management” on Page 7. Figures 1 and 2, and references have been modified to take into account this comment.
Reviewer 2 Report
In this review, Wils and colleagues provide an overview of the problem of visceral pain in IBD, optimally schematizing both the pathophysiological aspects and those relating to the management of this very important issue.
My only suggestion is to introduce a paragraph in the part concerning the treatment in which to summarize the most significant data relating to the possible use of probiotics in the treatment of visceral pain.
Author Response
Lille, July 2022, 14th
Dear Editors and Reviewers,
We are pleased with the interest you put into analyzing our paper entitled “Abdominal Pain in Inflammatory Bowel Diseases: a clinical challenge”, and we are thankful for the quality of the comments and the rapid processing, which have allowed us to improve the quality of our work. The issues that you identified were analyzed and changes in the writing have been done accordingly. Reviewer questions and concerns are addressed point by point in the following section. All manuscript changes resulting from reviewer suggestions are embedded in the paper and highlighted (in red color) as required.
Reply for reviewer 2
My only suggestion is to introduce a paragraph in the part concerning the treatment in which to summarize the most significant data relating to the possible use of probiotics in the treatment of visceral pain.
Agree. Restoring the microbial flora seems to be an interesting option in the management of IBS. Some probiotics have demonstrated their effectiveness in improving symptoms, including visceral pain for patients with IBS (for example L. Plantarum 299v, B. Bifidum MIMBb75). The use of probiotics in IBD could be a therapeutic option to improve dysbiosis, and consequently abdominal pain but requires further investigations.
This modification appears in the paragraph “Pain management” on Page 7.